# Increased Binding of von Willebrand Factor to Sub-Endothelial Collagen May Facilitate Thrombotic Events Complicating *Bothrops lanceolatus* Envenomation in Humans

**DOI:** 10.3390/toxins15070441

**Published:** 2023-07-03

**Authors:** Olivier Pierre-Louis, Dabor Resiere, Celia Alphonsine, Fabienne Dantin, Rishika Banydeen, Marie-Daniela Dubois, Hossein Mehdaoui, Remi Neviere

**Affiliations:** 1Cardiovascular Research Team EA7525, University of the French West Indies (Université des Antilles), 97233 Fort de France, France; uag.martinique@gmail.com (O.P.-L.); dabor.resiere@chu-martinique.fr (D.R.); alphonsine_celia@hotmail.com (C.A.); fabienne.dantin@chu-martinique.fr (F.D.); rishika.banydeen@chu-martinique.fr (R.B.); marie-danielaaymard.dubois@chu-martinique.fr (M.-D.D.); hossein.mehdaoui@chu-martinique.fr (H.M.); 2Department of Critical Care Medicine and Toxicology, University Hospital of Martinique (CHU Martinique), 97200 Fort-de-France, France; 3Department of Cardiology, University Hospital of Martinique (CHU Martinique), 97200 Fort-de-France, France

**Keywords:** *Bothrops* snake, *B. lanceolatus*, envenomation, von Willebrand factor, collagen binding activity

## Abstract

Consumption coagulopathy and hemorrhagic syndrome exacerbated by blood anticoagulability remain the most important causes of lethality associated with *Bothrops* snake envenomation. *Bothrops* venom also engages platelet aggregation on the injured endothelium via von Willebrand factor (vWF) interactions. Besides platelet aggregation, some *Bothrops* venom toxins may induce qualitative thrombopathy, which has been in part related to the inhibition of vWF activation. We tested whether *B. lanceolatus* venom impaired vWF to collagen(s) binding (vWF:CB) activity. Experiments were performed with *B. lanceolatus* crude venom, in the presence or absence of Bothrofav, a monospecific *B. lanceolatus* antivenom. Venom of *B. lanceolatus* fully inhibited vWF to collagen type I and III binding, suggesting venom interactions with the vWF A3 domain. In contrast, *B. lanceolatus* venom increased vWF to collagen type VI binding, suggesting the enhancement of vWF binding to collagen at the vWF A1 domain. Hence, *B. lanceolatus* venom exhibited contrasting in vitro effects in terms of the adhesive properties of vWF to collagen. On the other hand, the antivenom Bothrofav reversed the inhibitory effects of *B. lanceolatus* venom on vWF collagen binding activity. In light of the respective distribution of collagen type III and collagen type VI in perivascular connective tissue and the sub-endothelium, a putative association between an increase in vWF:CB activity for collagen type VI and the onset of thrombotic events in human *B. lanceolatus* envenomation might be considered.

## 1. Introduction

Features of human *Bothrops* spp. snake envenomation include marked local damage and, in some severe cases, abnormal hemostasis with systemic hemorrhage resulting from the synergistic action of several venom toxins [1,2]. The most toxic constituents of *Bothrops* spp. venom involved in impaired hemostasis and hemorrhage, include snake venom serine proteases (SVSPs), snake venom metalloproteinases (SVMPs), phospholipase A_2_ (PLA_2_), disintegrins (DIS), and C-type lectin-like proteins (CTL) [1]. Along with coagulation factor consumption and hemorrhagic activity, venom toxins can alter several platelet functions, including adhesion to the sub-endothelial extracellular matrix and aggregation, which may lead to initial clot or thrombus formation [3,4,5,6,7]. Of note, multistage processes that engage platelet aggregation have been described in *Bothrops* spp. envenomation [4].

Upon vascular injury, platelets initially adhere transiently to the sub-endothelial von Willebrand factor through interaction with the glycoprotein Ibα (GPIbα) platelet receptor. The von Willebrand factor (vWF) is a massive, shear-sensitive, homopolymeric protein that is critically involved in the initiation of platelet adhesion at high shear [8]. The shear-sensitive A1-A2-A3 region of vWF is important for platelet adhesion because this region contains the binding sites for glycoprotein Ibα and collagen structures [9,10]. In the extracellular matrix, collagen type I and collagen type III constitute the major part of the interstitial matrix [11,12,13]. Collagen type III is found extensively in connective tissues such as skin, lung, liver, intestine, and the perivascular connective tissue [12]. Collagen type III serves as a ligand for several proteins, such as the G-protein coupled receptor-56, von Willebrand factor, and α2β1 integrin [12]. Collagen type VI is a unique beaded filament collagen found in the interface between the basement membrane and interstitial matrix of many tissues, including the dermis, skeletal muscle, kidney, cornea, tendon, skin, cartilage, intervertebral discs, adipose tissue, and blood vessels [13]. Collagen type VI has a fundamental function anchoring endothelial basement membranes by interacting with collagen type IV, and also serves as a ligand for several proteins including biglycan, decorin, perlecan, neural/glial antigen 2 proteoglycan, fibronectin, tenascin, and α2β1 integrin [13].

Overall, vWF has a central role in primary hemostasis where it mediates platelet adhesion to damaged vascular sub-endothelium, and subsequently initiates platelet aggregation. Following a vascular injury, vWF binds specifically to fibrillar collagen. Binding sites for fibrillar collagen have been identified within vWF domains A1 and A3, although mutagenesis studies suggest that the major site in domain A3 and the minor site in domain A1 interact with different targets on collagen [9,10]. The vWF A3 domain is necessary and sufficient to support binding to fibrillar collagen types I and III, while the A1 domain is involved in binding to collagen type VI [9,10]. During platelet aggregation processes, vWF binds rapidly to exposed collagen vessel structures and enables platelet arrest from fast-flowing blood through the interaction of its A1 domain with the platelet GPIbα receptor. Rotational force imposed by flowing blood causes platelets to translocate over immobilized vWF until α2β integrin receptors engage their respective ligands and mediate permanent adhesion, spreading, and aggregation [8]. Collagen binding (vWF:CB) assays, usually performed using collagen(s) to capture vWF, determine quantitative binding [9,10]. These assays are initially diagnostic tests that quantify the binding capacity of vWF -A1 and -A3 domains to collagen (the main sub-endothelial matrix component), in order to improve the diagnosis and differentiation of the qualitative variant of von Willebrand disease, the most common inherited bleeding disorder [9,10].

Interactions between platelet function and snake venom constituents are multiple and complex. Thrombocytin, an SVSP from *B. atrox* venom, can induce platelet adhesion in vitro via calcium mobilization [14]. Bothrombin, an SVSP from *B. jararaca*, activates platelet aggregation in vitro by interacting with platelet GPIbα receptor in the presence of exogenous fibrinogen [15]. *Bothrops* PLA2 from *B. jararacussu,* such as bothropstoxin, is able to induce platelet aggregation through multiple signal transduction pathways, including thromboxane A_2_ formation and activation of protein kinase cascades [16]. Botrocetin, a CTL from *B. jararaca* venom, and aspercetin, a CTL from *B. asper* venom, can induce platelet aggregation in the presence of vWF, promoting its interaction with platelet GPIbα receptor [6,17,18,19]. Besides the activation of platelet aggregation, *Bothrops* spp. venom toxins may also induce qualitative thrombopathy [2]. For example, bothrasperin, a disintegrin from *B. asper* venom elicits platelet aggregation induced by collagen and ADP, thus altering the interaction of fibrinogen with the platelet integrin α2β3 receptor [20]. In human *B. jararaca* envenomation, platelets harvested from circulating blood display hypoaggregation to ristocetin and collagen [21]. Such antiaggregant properties have been attributed to the inhibitory effects of jararhagin, a P-III-type SVMP of *B. jararaca*, on vWF-to-collagen binding and its interaction with GPIb and the α2-subunit I domain of the platelet surface α2β integrin [22,23]. Atroxlysin-I and atroxlysin III (two SVMPs from *B. atrox)* and basparin A (an SVMP from *B. asper)* also inhibit collagen-dependent platelet aggregation independently of their proteolytic activities [24,25]. As a distinct mechanism, the proteolytic degradation of vWF by SVMPs [26,27,28], especially with the loss of high molecular weight bands and the increase in low molecular weight fragments, can further determine thrombopathy in *Bothrops* spp. envenomation [18,29,30].

In some cases of *Bothrops* spp. envenomation, platelet activation is associated with microthrombi formation, i.e., thrombotic microangiopathy [31,32,33,34,35,36]. The latter is usually observed in *Bothrops* spp. envenomation associated with consumption coagulopathy, which is typically marked by prolonged clotting times, clotting factor deficiencies (i.e., hypofibrinogenemia, low factor V, and low factor VIII), and elevated D-dimer [31,32,33,34,35,36]. Snakebite-associated thrombotic microangiopathy presents with red blood cell fragments (schistocytes) on the peripheral blood film and delayed phase thrombocytopenia [31,32,33,34,35,36]. Histologically, small vessel wall injury and micro-thrombosis are typically described and may lead to end-organ ischemia [31,32,33,34,35,36]. Multiple macro-thrombotic events, such as fatal pulmonary embolism, myocardial infarction, and cerebral ischemic stroke, have also been reported in human *B. lanceolatus* envenomation in the absence of thrombotic microangiopathy [37,38,39,40]. The pathogenesis of these thrombotic events still has to be fully elucidated. Proposed mechanisms include the switch of the endothelium to a prothrombotic phenotype with the overexpression of tissue factor, cytokines, and adhesion molecules, along with the activation of platelets, complement, and coagulation cascade [37,38,39,40].

The potential role of *B. lanceolatus* venom (frequent trigger of pro-thrombotic events) in inducing changes in vWF collagen binding (vWF:CB) activity has not been previously explored. The main objective of the present study was to test whether crude venom of *B. lanceolatus,* captured in the wild on the French Caribbean island of Martinique, induces in vitro changes in the adhesive properties of vWF on collagen(s), and whether the monospecific antivenom Bothrofav prevents venom-induced alterations of vWF:CB activities.

## 2. Results

In conditions of low vWF concentrations (vWF^low^) and high vWF concentrations (vWF^high^), vWF collagen type I binding (vWF:CB) activities were partially inhibited in the presence of *B. lanceolatus* venom at the concentration 1 µg/mL (Figure 1a,b). Venom concentrations above 1 µg/mL fully inhibited vWF collagen type I binding (vWF:CB) activities in vWF^low^ and vWF^high^ conditions (Figure 1a,b). Experiments with concentrations below 1 µg/mL were not performed.

vWF collagen type III binding (vWF:CB) activities in conditions of low vWF concentrations (vWF^low^) and high vWF concentrations (vWF^high^) were not inhibited in the presence of *B. lanceolatus* venom concentrations under 1 µg/mL (Figure 2a,b). Conversely, venom concentrations at 1 µg/mL and above fully inhibited vWF collagen type III binding (vWF:CB) activities in vWF^low^ and vWF^high^ conditions (Figure 2a,b).

In conditions of low vWF concentrations (vWF^low^), vWF collagen type VI binding (vWF:CB) activities were increased in the presence of *B. lanceolatus* venom at concentrations of 0.001, 0.01, and 0.1 µg/mL, while venom concentrations of 1 µg/mL and above had no effect. In conditions of high vWF concentrations (vWF^high^), venom concentrations of 0.001, 0.01, and 5 µg/mL were associated with increases in vWF collagen type VI binding (vWF:CB) activity (Figure 3a,b). 

The monospecific antivenom Bothrofav was able to fully reverse vWF collagen binding activities in conditions of *B. lanceolatus* venom concentrations of 1, 5, and 10 µg/mL, which inhibited vWF collagen type III and type I binding activities (vWF:CB) in both low vWF concentrations (vWF^low^) (Figure 4a,c) and high vWF concentrations (vWF^high^) (Figure 4b,d). At a 50 µg/mL *B. lanceolatus* venom concentration, the protective effects of Bothrofav were significantly suppressed (Figure 4).

Figure 5 displays the effects of varying concentrations of *B. lanceolatus* venom (0.1, 1, and 5 µg/mL) on vWF collagen type I, III, and VI binding activities (vWF:CB), expressed as percent of control in conditions of high vWF concentrations. Figure 5 highlights the contrasting effect of *B. lanceolatus* venom on collagen type I and collagen type III vWF binding properties versus collagen type VI. No comparisons between collagen type I, type III, and type VI vWF:CB were performed.

*B. lanceolatus* venom at the concentration of 1 µg/mL elicited a significant increase in vWF antigen (vWF:Ag) levels, while incubation with higher *B. lanceolatus* venom concentrations (10 and 50 µg/mL) was associated with reduced vWF:Ag levels (Figure 6).

## 3. Discussion

Despite initial local damage similar to that described after *B. atrox* snakebites, *B. lanceolatus* snakebites are infrequently responsible for systemic bleeding and anticoagulability. Instead, *B. lanceolatus* snakebites are commonly complicated by multiple systemic macro-thrombosis within 48 h after the bite, if no rapid antivenom administration is initiated [37,38,39,40]. The proposed mechanisms of these thrombotic events are related to the switch of the endothelium to a prothrombotic phenotype. The activation of vWF can play a critical role in these prothrombotic conditions by facilitating platelet tethering to injured sub-endothelium through binding sites for collagen(s) and the platelet GPIb receptor.

The potential role of *B. lanceolatus* venom in inducing changes in vWF collagen binding activity has not been previously investigated. In the present in vitro study, vWF to collagen type I and type III binding activities, which are determined by vWF A3 domain interactions, were fully inhibited by *B. lanceolatus* venom. In sharp contrast, *B. lanceolatus* venom increased vWF to collagen type VI binding, which is determined by vWF A1 domain interactions. Pre-incubation with the monospecific antivenom Bothrofav reversed the inhibitory effects of *B. lanceolatus* venom on vWF collagen type I and type III binding activities. Of note, we observed an increase in vWF:Ag levels in conditions of low *B. lanceolatus* venom concentrations, possibly due to vWF proteolysis. Overall, our results are consistent with previous studies showing that many components of *Bothrops* spp. venom interact with the vWF A1 and A3 domains for collagen binding, to regulate platelet adhesion and aggregation [3,5,7,27,28]. For example, C-type lectin-like proteins, botrocetin and bitiscetin, previously known as coagglutinins, induce platelet agglutination via the enhancement of vWF A1 domain binding sites [5,18,19,41]. Other molecular forms of bitiscetin, such as bitiscetin-2 and bitiscetin-3, interact with the vWF A3 domain binding site [42,43]. Jararhagin, a PIII SVMP containing a metalloproteinase domain followed by disintegrin-like and cysteine-rich domains, interacts with the vWF A1 domain [27,28,30]. Atroxlysins (P-I and P-III hemorrhagic metalloproteinases) isolated from *B. atrox*, can dose-dependently inhibit ADP- or collagen-triggered platelet aggregation via the vWF A1 domain [25,44].

In our study, we found that *B. lanceolatus* venom fully inhibited vWF to collagen type I and type III binding. This is consistent with previous studies indicating that interactions between platelets, vWF, and collagen are reduced, owing to the direct inhibitory properties of *Bothrops* spp. venom on glycoprotein Ib (GPIb) and α2β integrin platelet receptors [18,22,23,44,45,46,47]. In line, human victims of *B. jararaca* envenomation, who present systemic bleeding, display vWF collagen binding inhibition as evidenced by hypoaggregation to ADP, ristocetin, and collagen [21]. Interestingly, we found that the monospecific antivenom Bothrofav was able to fully reverse vWF-collagen binding activity (vWF:CB) inhibition induced by *B. lanceolatus* venom. Indeed, the Bothrofav antivenom was able to fully reverse vWF collagen type I and type III binding inhibition. The effects of Bothrofav on vWF:CB inhibition was obtained using a co-incubation procedure with *B. lanceolatus* venom. The procedure is the gold standard protocol to test antivenom efficacy in preclinical studies. Previous clinical studies have shown that antivenom therapy was able to partially restore platelet hypoaggregation for ADP and ristocetin in *Bothrops* spp. envenomation [21]. Thereon, further studies are required to address whether Bothrofav is able to reverse vWF:CB inhibition in human venom envenomation by *B. lanceolatus*.

Nonspecific proteolytical vWF degradation by different venom enzymes can also indirectly contribute the to vWF-to-collagen binding deficit [26,27,28,29,30]. For example, the inhibition of ristocetin-induced aggregation by jararhagin has been attributed to a direct effect on vWF, rather than to its action on the GP Ib-receptors [23]. In our study, the effects of *B. lanceolatus* venom on vWF:Ag levels were dose-dependent. Low venom concentration (1 µg/mL) increased vWF:Ag levels, which might be related to the proteolytic action of *B. lanceolatus* venom leading to newly exposed epitopes in cleaved vWF molecules, thereby amplifying the final signal as detected by the ELISA technique [28,30]. This has already been observed in *B. jararaca* envenomation in rodents, where, despite an observed general decrease in vWF:Ag levels in the overall rodent sample, some animal subjects nevertheless displayed elevated plasma vWF antigen [18]. Proteolytical vWF degradation may be due to the large predominance of SVMPs in *B. lanceolatus* venom (74%) [48,49]. In contrast, high venom concentrations (10 and 50 µg/mL) reduced vWF:Ag levels, which may suggest vWF conformation changes in terms of multimerization and masking epitopes, thereby reducing the final signal as detected by the ELISA technique. The effects of *B. lanceolatus* venom on ADAMTS13 activity and the accumulation of ultra high molecular weight multimers of vWF need to be further explored.

In sharp contrast with its inhibitory effect on vWF to collagen type I and type III binding, *B. lanceolatus* venom enhanced vWF to collagen type VI binding activity. This result is consistent with studies showing that botrocetin and bitiscetin, two venom C-type lectin-like proteins, induce platelet agglutination via the enhancement of vWF to collagen binding site affinity via the A1 domain [5,19,41]. The presence of C-type lectin-like protein(s) in *B. lanceolatus* venom has been demonstrated by proteomic studies, while the precise identity of these proteins is not yet known [48]. The respective roles of vWF A1 and A3 domains and their interactions in collagen binding processes have been extensively investigated. In static conditions, vWF with deleted A3 domain binds poorly to collagen fibrils, in comparison to vWF with a deleted A1 domain and intact VWF, which both bind equally well to collagen [46,47,48,49]. In flow conditions, platelet adhesion to collagen was only observed when vWF was allowed to first bind to collagen through the A3 domain, then subsequently to the A1 domain responsible for platelet recruitment via the GP Ibα receptor [50,51,52,53]. In line with the prominent role of the vWF A1 domain in platelet aggregation via the GP Ibα receptor and its binding to collagen VI of the sub-endothelium, we propose that increased vWF to collagen type VI binding could be a major pathophysiological mechanism leading to multiple thrombotic events described in *B. lanceolatus* envenomation in humans. Indeed, while the localization of collagen types I and III in the connective tissues, below the interfacing zone between the sub-endothelial basement membrane and the interstitium, raises some doubts about their physiological relevance in the early phases of vascular repair, the intimate association of collagen type VI with the basement membrane strongly supports an important role for this collagen type in binding circulating vWF at the site of vascular injury [54,55,56]. Furthermore, the co-localization of collagen type VI and vWF in the vascular sub-endothelium also supports that collagen type VI may represent a central component of the sub-endothelial matrix contributing to platelet aggregation upon rupture of the blood vessel wall [54,55,56]. Increased vWF to collagen binding has been previously reported in pathological conditions, such as inflammation related to infection and cardiovascular diseases (coronary heart diseases and cerebral stroke) [57,58,59]. Hence, it may be hypothesized that the macro-thrombotic events associated with human *B. lanceolatus* envenomation [37,38,39,40] might be related at least in part to increased vWF to collagen type VI binding.

In our study, Bothrofav, a monospecific antivenom against *B. lanceolatus*, was able to prevent the inhibitory effects of *B. lanceolatus* venom on vWF collagen type I and type III binding activities. Since 1991, a highly purified and monospecific antivenom against *B. lanceolatus* has been manufactured by Aventis-Pasteur (Lyon, France) as Bothrofav [60]. This antivenom is an F(ab′)2 preparation obtained by pepsin digestion and ammonium sulfate fractionation of hyperimmune plasma from horses immunized with the venom of *B. lanceolatus*. Bothrofav is highly effective in reducing mortality and preventing the development of thrombosis and other systemic disturbances in these envenomations [60,61]. The preclinical assessment of the neutralizing capacity of Bothrofav has corroborated its efficacy in the inhibition of lethal activities, and, most importantly, of toxic and enzymatic activities of the *B. lanceolatus* venom [62]. Third-generation antivenomics analysis has further proven Bothrofav’s high neutralizing efficacy of all the major immunogenic protein components of *B. lanceolatus* venom (i.e., metalloproteinases, phospholipases A_2_, serine proteinases, C-type lectin-like proteins, and disintegrins), with the exception of poor immunogenic peptides [63]. In our study, the Bothrovav-induced prevention of the inhibitory effects of *B. lanceolatus* venom on vWF collagen type I and type III binding may be related to the reported beneficial effects of the antivenom on coagulopathic activities [64,65].

## 4. Conclusions

Our in vitro study suggests that *B. lanceolatus* venom alters the adhesive properties of vWF on collagen and increases vWF:Ag levels, possibly due to vWF cleavage. *B. lanceolatus* venom seemed to fully inhibit vWF to collagen type I and type III binding, which was reversed by the monospecific antivenom Bothrofav. In contrast, *B. lanceolatus* venom increased vWF to collagen type VI binding. In light of the respective distribution of collagen type III and collagen type VI in perivascular connective tissue and the sub-endothelium, a plausible association between increased vWF:CB activity for collagen type VI and the onset of thrombotic events in human *B. lanceolatus* envenomation might be considered.

## 5. Material and Methods

### 5.1. Venom and Antivenom

For venom collection, we milked the front fangs of snakes. The animals were forced to bite through a thin membrane and release their venom into a clean glass vessel. The crude venom was obtained from adult wild *B. lanceolatus* specimens, captured in Martinique. Venom samples were pooled from the milking of five specimens. All venom samples were lyophilized (Freezone, Labconco, Kansas City, MO, USA) and stored at −80 °C until use (stock solution 10 mg/mL). *B. lanceolatus* venom was used in experiments at the final concentrations of 1, 5, 10, and 50 µg/mL. Bothrofav is a preparation containing F(ab’)2 fragments that have the property of neutralizing *Bothrops lanceolatus* venom. These equine F(ab’)2 fragments ligate venom antigens present in circulating blood to form inactive F(ab’)2-antigen complexes, in turn reducing the amount of free venom in circulation. The monospecific anti-venom Bothrofav (Sanofi Pasteur, Lyon, France) was used (batch J8216; protein concentration of 20.7 ± 0.05 g/dL). Bothrofav was used in experiments at the final concentrations of 0.21 mg/mL and 0.41 mg/mL.

### 5.2. Control Plasma and von Willebrand Factor

Lyophilized control plasmas with “low vWF level” (0.36 UI/mL, vWF^low^) or “high level” (1.23 UI/mL, vWF^high^) included in ELISA assay kits were used for vWF:Ag and vWF:CB experiments (Technozym ELISA vWF:Ag and vWF:CB, Cryopep, Montpellier, France). For vWF multimer experiments, Cryocheck Pooled Normal Plasma and a highly purified plasma-derived von Willebrand factor Wilfactin, almost devoid of FVIII and containing high molecular weight (HMW) multimers with a distribution similar to normal plasma, were respectively purchased from Cryopep (Montpellier, France) and LFB-Biopharmaceuticals (Les Ulis, France).

### 5.3. Determination of vWF Antigen

vWF Ag ELISA kit was used for the quantitative determination of human von Willebrand factor (vWF) concentrations (Technozym ELISA assay kit, Cryopep, Montpellier, France) according to the manufacturer’s instructions. This assay employs the quantitative sandwich enzyme immunoassay technique utilizing a polyclonal anti-vWF Ag antibody and a vWF Ag-HRP conjugate. The assay sample and buffer were incubated together with vWF Ag-HRP conjugate in the pre-coated plate for one hour. After the incubation period, the wells were decanted and washed five times. The wells were then incubated with a substrate for HRP enzyme. The product of the enzyme-substrate reaction formed a blue-colored complex. Finally, a stop solution was added to stop the reaction, which then turned the solution yellow. Color intensity was measured spectrophotometrically at 450 nm in a microplate reader. Color intensity was inversely proportional to vWF Ag concentration, since vWF Ag from samples and vWF Ag-HRP conjugate compete for the anti-vWF Ag antibody binding site. As the number of sites is limited, the more sites that are occupied by vWF Ag from the sample, the fewer the sites that are free to bind the vWF Ag-HRP conjugate. A standard curve was plotted representing the color intensity (O.D.) according to the concentration of standards. The vWF Ag concentration in each sample was interpolated from this standard curve.

### 5.4. Determination of vWF Collagen Binding (Activity)

The vWF collagen binding assay (vWF:CB) was performed with an indirect ELISA, which involved two binding processes of primary antibody and labeled secondary antibody. vWF in patient plasma was captured in microtiter plates coated with human collagen(s). Unbound material was then washed away and a solution of antibody to human vWF, conjugated to an enzyme, was added to ‘tag’ onto any captured vWF. Unbound conjugate was washed off and a substrate for the enzyme was added, the product of the enzyme-substrate reaction being colored. Color intensity was in direct proportion to the degree of conjugate binding, itself proportional to the amount of vWF captured, and thus to vWF collagen binding activity. A standard curve was constructed from a pool of normal plasma donors. Bound vWF was detected using polyclonal anti-human vWF antibodies (anti-vWF-POX) labeled with horse radish peroxidase (HRP). Coloring reaction was performed with orthophenylenediamine and hydrogen peroxide. The reaction was stopped with sulfuric acid. Absorbance was measured at 450 nm. Specific vWF collagen binding assay kits were chosen for evaluating the functional activity of vWF, i.e., the interaction between vWF and the matrix collagen I, collagen III, and collagen VI (Technozym ELISA assay kit collagen type I, III, and VI purchased from Cryopep (Montpellier, France)).

### 5.5. Incubation Protocol

Microtiter plates coated with human collagen(s) were incubated with either phosphate saline buffer (PBS), *B. lanceolatus* venom in PBS, or a mixture of *B. lanceoltaus* venom plus Bothrofav in PBS at the indicated concentrations. Binding activity of vWF to collagen was studied after 45 min incubation of plasma samples with either low vWF level (vWF^low^) or high vWF level (vWF^high^) included in ELISA assay kits.

### 5.6. Statistical Analysis

Quantitative data were presented as mean ± standard deviation (SD). The Shapiro–Wilk test was used to test for normal distribution of quantitative data. Data were analyzed by using analysis of variance ANOVA. When a significant difference was found, we identified specific differences between groups with a sequentially rejective Bonferroni procedure. After application of the Bonferroni correction, the level of statistical significance was set at *p* < 0.05. Results were analyzed with the Prism 6 for Windows software (Graphpad, Boston, MA, USA).

## Figures and Tables

**Figure 1 toxins-15-00441-f001:**
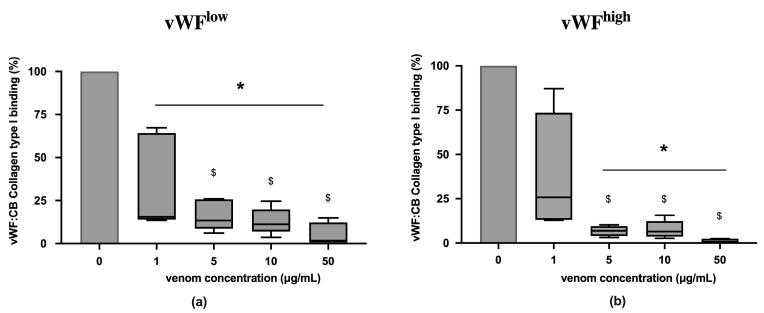
(**a**) Effects of B. lanceolatus venom on vWF collagen type I binding activity (vWF:CB collagen type I binding), expressed as percent of control, in conditions of low vWF concentrations (vWF^low^); (**b**) Effects of B. lanceolatus venom on vWF collagen type I binding activity (vWF:CB collagen type I binding) in conditions of high vWF concentrations (vWF^high^). Data are displayed as box plots (median, whiskers min–max) of 5–8 independent experiments. * indicates statistical difference (*p* < 0.05) with control (phosphate saline buffer, PBS). $ indicates statistical difference (*p* < 0.05) with venom concentration of 1 µg/mL.

**Figure 2 toxins-15-00441-f002:**
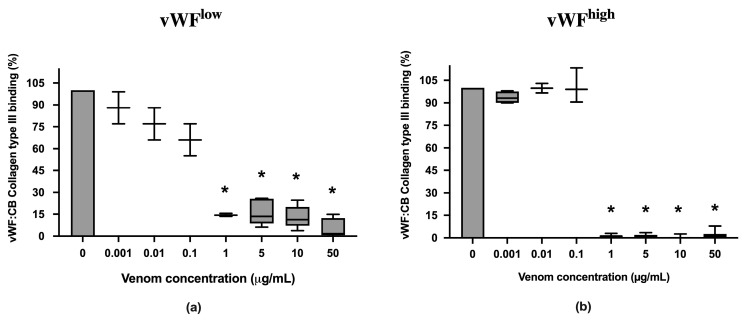
(**a**) Effects of *B. lanceolatus* venom on vWF collagen type III binding activity (vWF:CB collagen type III binding), expressed as percent of control, in conditions of low vWF concentrations (vWF^low^); (**b**) Effects of *B. lanceolatus* venom on vWF collagen type III binding activity (vWF:CB collagen type III binding) expressed as percent of control in conditions of high vWF concentrations (vWF^high^). Data are displayed as box plots (median, whiskers min–max) of 5–8 independent experiments. * indicates statistical difference (*p* < 0.05) with control (phosphate saline buffer, PBS).

**Figure 3 toxins-15-00441-f003:**
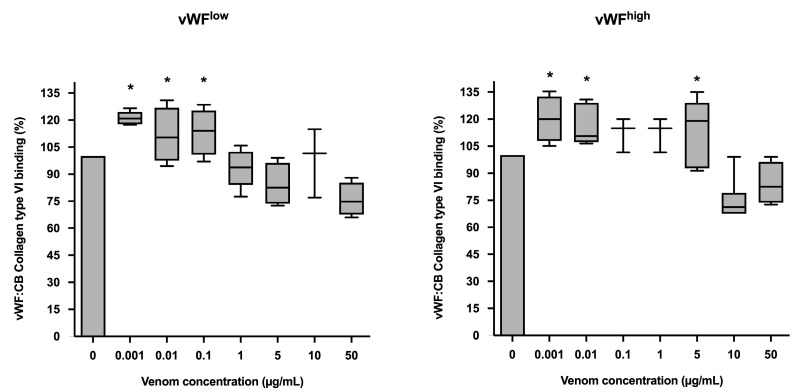
(**a**) Effects of *B. lanceolatus* venom on vWF collagen type VI binding activity (vWF:CB collagen type VI binding), expressed as percent of control, in conditions of low vWF concentrations (vWF^low^); (**b**) Effects of *B. lanceolatus* venom on vWF collagen type VI binding activity (vWF:CB collagen type VI binding) expressed as percent of control in conditions of high vWF concentrations (vWF^high^). Data are displayed as box plots (median, whiskers min–max) of 5–8 independent experiments. * indicates statistical difference (*p* < 0.05) with control (phosphate saline buffer, PBS).

**Figure 4 toxins-15-00441-f004:**
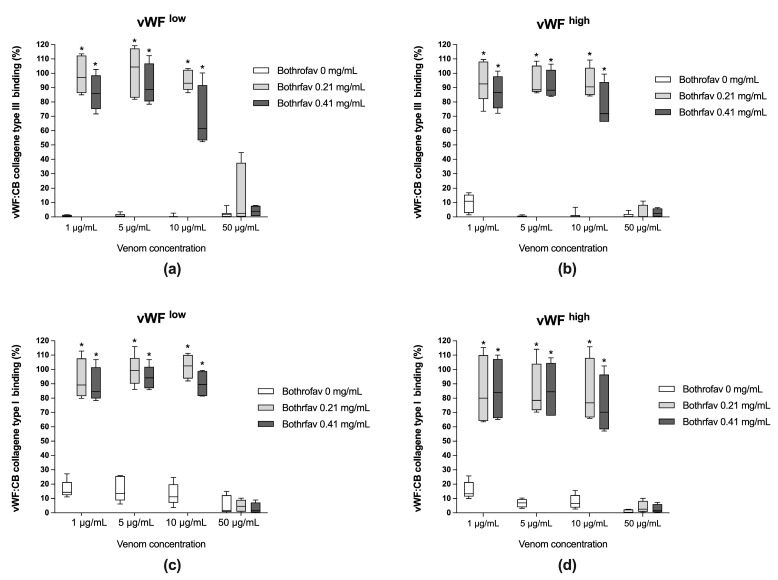
(**a**,**b**) Effects of Bothrofav antivenom on the inhibitory action of different *B. lanceolatus* venom concentrations (1, 5, 10, and 50 µg/mL) on vWF collagen type III binding activity (vWF:CB collagen type III binding), expressed as percent of control in conditions of low vWF concentrations (vWF^low^) and high vWF concentrations (vWF^high^); (**c**,**d**) Effects of Bothrofav antivenom on the inhibitory action of different *B. lanceolatus* venom concentrations (1, 5, 10, and 50 µg/mL) on vWF collagen type I binding activity (vWF:CB collagen type I binding), expressed as percent of control, in conditions of low vWF concentrations (vWF^low^) and high vWF concentrations (vWF^high^). Data are displayed as box plots (median, whiskers min–max) of 5–8 independent experiments. * indicates statistical difference (*p* < 0.05) with control (Bothrofav antivenom phosphate saline buffer vehicle).

**Figure 5 toxins-15-00441-f005:**
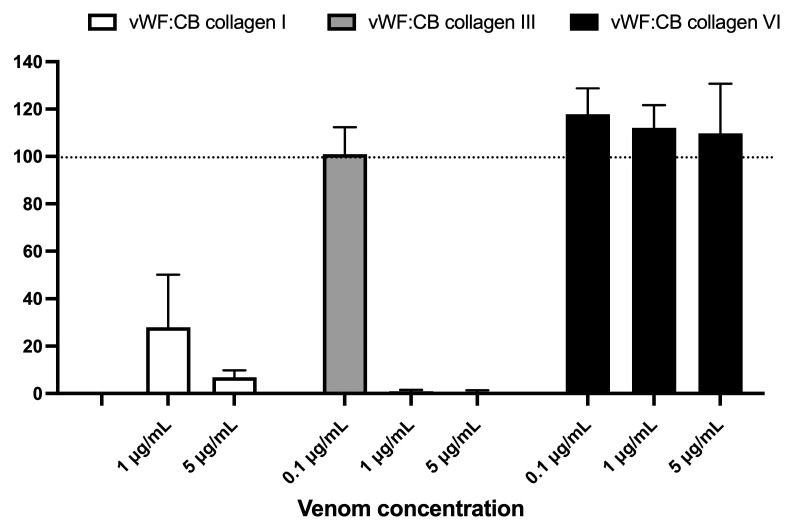
Effects of varying concentrations of *B. lanceolatus* venom (0.1, 1, and 5 µg/mL) on vWF collagen type I, III, and VI binding activities (vWF:CB collagen binding), expressed as percent of control without venom in conditions of high vWF concentrations (vWF^high^). Data are displayed as box plots (median, whiskers min–max) of 5–8 independent experiments.

**Figure 6 toxins-15-00441-f006:**
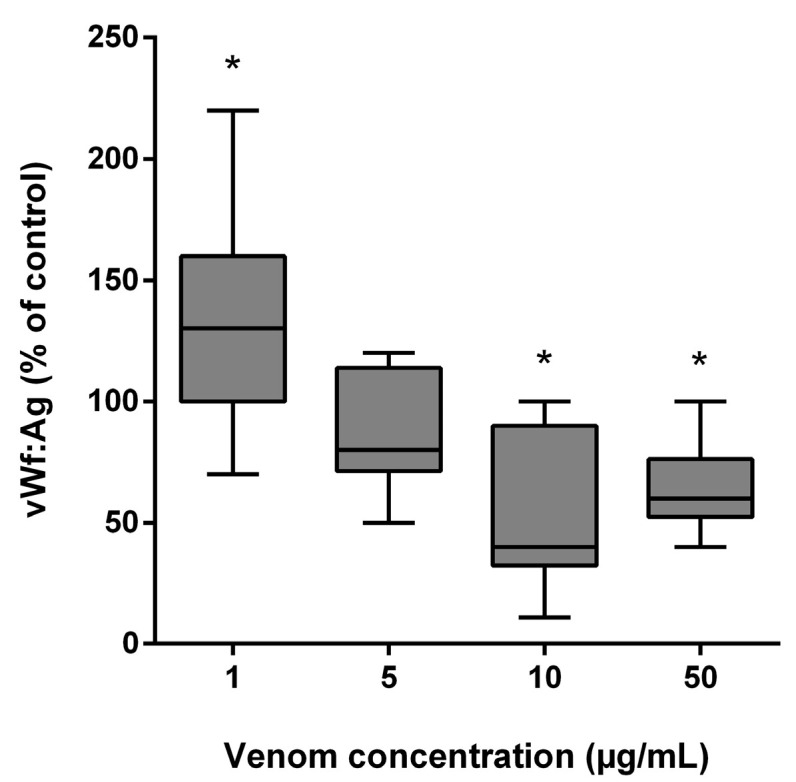
Effects of *B. lanceolatus* venom on vWF:Ag levels. Data are displayed as box plots (median, whiskers min–max) of 5 independent experiments. * indicates statistical difference (*p* < 0.05) with control.

## Data Availability

Data supporting reported results can be provided on request to the corresponding author.

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
