# Peer review of "Increased Binding of von Willebrand Factor to Sub-Endothelial Collagen May Facilitate Thrombotic Events Complicating Bothrops lanceolatus Envenomation in Humans"

_toxins, 2023, doi:10.3390/toxins15070441_

Round 1

Reviewer 1 Report

The authors report the activity of Bothrops lanceolatus venom on the VWF collagen binding activity assay. A putative association between the increase in VWF:CB activity for collagen type “VI” and thrombotic events in B. lanceolatus envenomation is claimed.

The paper is not very sound. In fact, I wonder if the B. lanceolatus venom concentrations used to study VWF:CB activity echo the venom concentrations found in the plasma patients bitten by Bothrops snakes. Usually, levels of ng/ml are found in patients, and the concentrations used in the paper overestimates the activity of the venom on collagen and VWF. Although very high venom concentrations induced a decrease in the binding of collagens I and III to VWF, concentrations below 1 µg/mL should have been used.

As far as I know, collagen type VI is found in skeletal muscle. I guess that authors used collagen type IV. I suggest the authors review it throughout the text.

Lines 18 -20 and 243-245– the sentences are unintelligible.

Lines 38 -39 – Is reference 8 about Bothrops venom?

Lines 47-48 – it is collagen type IV and not VI.

Review of the literature is really poor and information is wrong. Thrombocytin (ref. 10) nor bothropstoxin do not interact with GPIbalfa or VWF.

Figure 4 – where is the control in figure 4?

Line 179 – please define platelet agglutination? Do these reference deal with platelet agglutination?

Material and methods: detail how and how long venom samples were incubated with plasma and venom. Did collagen were incubated with the venom? Electrophoretic analyses of VWF or collagens incubated with the venom are essential to the carried out.

The text should be reviewed by a native English speaker.

Author Response

REVIEWER #1

The authors report the activity of Bothrops lanceolatus venom on the VWF collagen binding activity assay. A putative association between the increase in vWF:CB activity for collagen type “VI” and thrombotic events in B. lanceolatus envenomation is claimed. The paper is not very sound. In fact, I wonder if the B. lanceolatus venom concentrations used to study vWF:CB activity echo the venom concentrations found in the plasma patients bitten by Bothrops snakes. Usually, levels of ng/ml are found in patients, and the concentrations used in the paper overestimates the activity of the venom on collagen and VWF. Although very high venom concentrations induced a decrease in the binding of collagens I and III to VWF, concentrations below 1 µg/mL should have been used.

Authors: We thank the reviewer for its valuable comments. We agree that venom plasma concentrations in patients bitten by Bothrops snakes are found around 10 to 100-1000 ng/mL. Of note, it should be indicated that venom concentrations in the injured tissue may be far above. As required, we have performed new series of experiments aimed to evaluate the effects of venom concentrations below 1 µg/mL. Please find this new information in “Results” section and Figures of the revised manuscript for vWF:CB to collagen type  III and VI.

As far as I know, collagen type VI is found in skeletal muscle. I guess that authors used collagen type IV. I suggest the authors review it throughout the text.

Authors: The reviewer is right but some collagen type VI is found in the interface between the basement membrane and interstitial matrix of many tissues including the dermis, skeletal muscle, kidney, cornea, tendon, skin, cartilage, intervertebral discs, adipose tissue and blood vessels. Also, collagen type VI has a fundamental function anchoring endothelial basement membranes by interacting with collagen type IV. Please find comments on this specific issue in the “Introduction” section of the revised manuscript.

Lines 18 -20 and 243-245– the sentences are unintelligible.

Authors: Our aim was to report the activity of Bothrops lanceolatus venom on the VWF collagen type I, type III and type VI binding activity assay. We found that Bothrops lanceolatus venom inhibited binding of vWF to collagen type I and collagen type III, while Bothrops lanceolatus venom has an opposite effect on vWF to collagen type VI. Owing the respective distribution of collagen type III and type VI in perivascular connective tissue and the sub-endothelium, a putative association between the increase in VWF:CB activity for collagen type VI and thrombotic events in B. lanceolatus envenomation was hypothesized. Please find a better explanation of this rationale in the “Abstract” and “Discussion” sections of the revised manuscript.

Lines 38 -39 – Is reference 8 about Bothrops venom?

Authors: Indeed, Ref. 8 does not describe the well-described multistage processes involved in platelet aggregation in Botrops envenomation. According to reviewer’ comments, our confusing sentence has been modified in the revised manuscript. Please read “Multistage processes that engage platelet aggregation have been described in Bothrops spp envenomation [4]”.

Lines 47-48 – it is collagen type IV and not VI.

Authors: As state above, it is collagen type VI. Please find comments on this specific issue in the “Introduction” section of the revised manuscript.

Review of the literature is really poor and information is wrong. Thrombocytin (ref. 10) nor bothropstoxin do not interact with GPIbalfa or VWF.

Authors: We apologize for these confusing references and errors. Indeed, thrombocytin, a SVSP isolated from the venom of B. atrox induce platelet aggregation via calcium mobilization and is independent of GPIb-alpha or VWF interactions (our reference 10). Likewise, the ability of bothropstoxin to induce platelet activation has been related to the capacity to hydrolyze phophatidylcholine and the liberation of arachidonic acid (our reference 12), not with GPIb-alpha or VWF interactions. Please read the following sentences in the revised manuscript “Thrombocytin, a SVSP from B. atrox venom, can induce platelet adhesion in vitro via calcium mobilization [10]. Bothrombin, a SVSP from B. jararaca can induce platelet ad-hesion in vitro via platelet GPIbα receptor interaction in the presence of exogenous fi-brinogen [11]. Bothropstoxin, a PLA2 from B. jararacussu induces platelet aggregation through multiple signal transduction pathways, including thromboxane A2 formation and activation of protein kinase cascades [12].”

Figure 4 – where is the control in figure 4?

Authors: Please understand that Figure 4 displays B. lanceolatus venom (5µg/mL) effects on vWF collagen type I, III and VI binding activities (vWF:CB collagen binding), expressed as percent of control without venom in conditions of high vWF concentrations. Our aim was to highlight the contrasting effect of venom on collagen type I and collagen type III versus collagen type VI vWF binding properties. No comparisons between collagen type I, type III and type VI were performed.

Line 179 – please define platelet agglutination? Do these reference deal with platelet agglutination?

Authors: We apologize for this confusing wording. Please read “platelet adhesion and aggregation” in the revised manuscript.

Material and methods: detail how and how long venom samples were incubated with plasma and venom. Did collagen were incubated with the venom? Electrophoretic analyses of VWF or collagens incubated with the venom are essential to the carried out.

Authors: Details on how and how long venom samples were incubated with plasma and venom have provided in the revised “Materials and Methods” section of the revised manuscript

Comments on the Quality of English Language: The text should be reviewed by a native English speaker

Authors: The revised manuscript has been reviewed by a native English speaker.

Reviewer 2 Report

The submitted manuscript evaluates the effect of Bothrops lanceolatus venom on vWF to collagen binding activity. The main finding consists in the observation that the increased vWF-collagen type VI binding might be responsible for some of the pathological effect of B. lanceolatus envenomation.

The manuscript is logically constructed, with the results clearly presented. There are some minor errors that should be corrected, as follows:

·       Title: The complete name of the species should be given.

·       Throughout the text: Kindly format the text so that a space is inserted between values and units.

·       Line 12: “monospecific” typo

·       Line 18: “increased binding of vWF to collagen” or a similar correction of the phrase is required.

·       Line 96: “Bothrofav would prevent” or a similar correction is needed.

·       Line 111-113: Should be rephrased. As 1 µg/mL had only a partial inhibitory effect, the statement that “all concentrations used” inhibited vWF binding is incorrect.

·       Line 130: Effect of Bothrofav at 5 and 50 µg/mL is presented. Was the 10 µg/mL venom concentration tested as well? If yes, please include the results here. If not, why did the authors skip this concentration?

·       Line 151-153, Line 175-176: There is a contradiction in the text: “while incubation with lower B. lanceolatus venom concentrations (1 and 5 μg/mL) was associated with reduced vWF:Ag levels” and “Of note, we observed an increase of vWF:Ag levels in conditions of low B. lanceolatus venom concentrations, possibly due to vWF proteolysis.” Figure 5 supports the second sentence. Please revise and correct accordingly.

·       Line 153: vWF:Ag notation should be explained at first appearance.

·       Figure 5 caption: “B. lanceolatus” typo

·       Line 193: “vWF’ typo

·       Line 247: “For venom collection” typo

·       References: The references used are rather outdated. Probably less than half of them have been published in the last decade. Outdated bibliography usually represents either a topic of little relevance or a superficial review of the literature. Physiological and pathophysiologic mechanisms tend to be revaluated if new information becomes available, thus the authors are recommended to reevaluate the References list, possibly omitting outdated sources and introducing the latest advancements  in the proposed field.

Minor editing of English language required. 

Author Response

REVIEWER #2

The submitted manuscript evaluates the effect of Bothrops lanceolatus venom on vWF to collagen binding activity. The main finding consists in the observation that the increased vWF-collagen type VI binding might be responsible for some of the pathological effect of B. lanceolatus envenomation.

The manuscript is logically constructed, with the results clearly presented. There are some minor errors that should be corrected, as follows:

Authors: We thank the reviewer for its valuable comments.

Title: The complete name of the species should be given.

Authors: The complete name of the species has been given in the title of the revised manuscript.

Throughout the text: Kindly format the text so that a space is inserted between values and units.

Authors: A space has been inserted between values and units throughout the text of the revised manuscript.

Line 12: “monospecific” typo

Authors: Typo has been corrected.

Line 18: “increased binding of vWF to collagen” or a similar correction of the phrase is required.

Authors: Typo has been corrected.

Line 96: “Bothrofav would prevent” or a similar correction is needed.

Authors: Typo has been corrected.

Line 111-113: Should be rephrased. As 1 µg/mL had only a partial inhibitory effect, the statement that “all concentrations used” inhibited vWF binding is incorrect.

Authors: The reviewer is right. We apologize for this confusing wording. Please read an adequate presentation of the results of effects of B. lanceolatus venom on vWF collagen type I binding activity in the revised manuscript.

Line 130: Effect of Bothrofav at 5 and 50 µg/mL is presented. Was the 10 µg/mL venom concentration tested as well? If yes, please include the results here. If not, why did the authors skip this concentration?

Authors: Thanks to reviewer’ comment, we have included a new series of experiments using B. lanceolatus venom at concentrations of 1, 5, 10 and 50 µg/mL, which are displayed Figure 3 of the revised manuscript. Indeed, the monospecific antivenom Bothrofav was able to fully reverse vWF collagen binding activities in conditions of B. lanceolatus venom concentrations of 1, 5, and 10 µg/mL. Protective effects of Bothrofav were largely uncompleted for a 50 µg/mL B. lanceolatus venom concentration.

Line 151-153, Line 175-176: There is a contradiction in the text: “while incubation with lower B. lanceolatus venom concentrations (1 and 5 μg/mL) was associated with reduced vWF:Ag levels” and “Of note, we observed an increase of vWF:Ag levels in conditions of low B. lanceolatus venom concentrations, possibly due to vWF proteolysis.” Figure 5 supports the second sentence. Please revise and correct accordingly.

Authors: The reviewer is right. We apologize for this confusing wording. We meant that low B. lanceolatus venom concentration (1 μg/mL) was associated with increased vWF:Ag levels, while higher B. lanceolatus venom concentrations (10 and 50 μg/mL) were associated with reduced vWF:Ag levels.

Line 153: vWF:Ag notation should be explained at first appearance.

Authors: vWF:Ag notation has been explained at first appearance in the revised manuscript.

Figure 5 caption: “B. lanceolatus” typo

Authors: Typo has been corrected.

Line 193: “vWF’ typo

Authors: Typo has been corrected.

Line 247: “For venom collection” typo

Authors: Typo has been corrected.

References: The references used are rather outdated. Probably less than half of them have been published in the last decade. Outdated bibliography usually represents either a topic of little relevance or a superficial review of the literature. Physiological and pathophysiologic mechanisms tend to be revaluated if new information becomes available, thus the authors are recommended to reevaluate the References list, possibly omitting outdated sources and introducing the latest advancements in the proposed field.

Authors: Reference list has been updated where possible.

Comments on the Quality of English Language: Minor editing of English language required. 

Authors: The revised manuscript has been reviewed by a native English speaker.

Reviewer 3 Report

The article named “Increased von Willebrand factor to sub-endothelial collagen type VI binding may facilitate thrombotic events complicating B. lanceolatus envenomation in humans” shows new information about the effect on hemostasis from a bothropic snake venom (B. lanceolatus) by interaction with vWF. They proposed that the B. lanceolatus venom can inhibit vWF to collagen type I and III binding, suggesting its interactions with vWF A3 domain; and an increased binding of vWF to collagen type VI, suggesting enhancement at vWF A1 domain. They also demonstrated that the antivenom used for snake treatment (Bothrofav) reversed the inhibitory effects of venom on vWF collagen binding activity. Thus, the binding capacity of vWF to collagen type VI could be a critical pathophysiologic mechanism of multiple thrombotic events on B. lanceolatusenvenomation.

I suggest some improvement for the final version of paper:

·      Please, rewrite the paragraph from line 55 – 74; there are word repetitions that must be correct to improve the grammar quality.

·      Figure 1: Please describe what was used as a control for this experiment.

·      Effects of B. lanceolatus venom on vWF collagen (line 98): Suggestion for inclusion new results of more concentrations lower than used on it to determine the value for 50% of activity. 

·      Figure 2: As suggestion the author could perform the experiment of B. lanceolatus effects on vWF collagen binding (high and low concentration of vWF) including venom concentration between 1 and 5 µg/mL to cover the gap of activity showed on it; it will be possible to go for no/less activity to 100% of activity.

·      Figure 3:  Please, improve the visual quality of this figure.

·      It is known that the conventional treatment for snake envenommation is performed according to the severity of bite (mild, moderate and severe). The author showed on Figure 3 two different concentrations of antivenom, which were able to inhibit the venom effect at 5µg/ml. However, they showed that these concentrations were inefficient when used the concentration 50µg/ml of venom. As suggestion, the authors should explore the effect difference according to venom concentration and connect the result obtained with the clinical use of antivenom, describing what is the usual dosage used for clinical treatment and efficiency percent showed on the paper. Thus, I highly suggest testing and show more than two concentration of snake venom for the inhibition assay with antivenom.

·      Grammar revision. There are many long sentence and word repetition on it.

·      The authors should explore more information about the Bothrifav’s composition and the possible mechanism for venom inhibition, especially for the hemorrhagic effect.

·      Sentence from line 271 – 273: please do a grammar revision, there are some incorrect verb conjugations on it.

·      Item “Determination of vWF collagen binding activity assay”: please check how the world vWF:CB was write, there are two different way for them.

In general, the paper showed clearly an important aspect for Bothrops venom and will contribute for giving more information for futures studies on Toxinology. In conclusion, the improvement suggested will increase the paper quality for publication.

In general the English language is clearly, however must be performed an review to improve the quality for publication.

Author Response

REVIEWER #3

The article named “Increased von Willebrand factor to sub-endothelial collagen type VI binding may facilitate thrombotic events complicating B. lanceolatus envenomation in humans” shows new information about the effect on hemostasis from a bothropic snake venom (B. lanceolatus) by interaction with vWF. They proposed that the B. lanceolatus venom can inhibit vWF to collagen type I and III binding, suggesting its interactions with vWF A3 domain; and an increased binding of vWF to collagen type VI, suggesting enhancement at vWF A1 domain. They also demonstrated that the antivenom used for snake treatment (Bothrofav) reversed the inhibitory effects of venom on vWF collagen binding activity. Thus, the binding capacity of vWF to collagen type VI could be a critical pathophysiologic mechanism of multiple thrombotic events on B. lanceolatus envenomation.

I suggest some improvement for the final version of paper:

Please, rewrite the paragraph from line 55 – 74; there are word repetitions that must be correct to improve the grammar quality.

Authors: The paragraph has been improved according to reviewer’ comments. Please read the fourth paragraph of the revised manuscript.

Figure 1: Please describe what was used as a control for this experiment.

Authors: Thank to reviewer’ comments, we have included a paragraph entitled “Incubation protocol” in the “Materials and Methods” of the revised manuscript. Phosphate saline buffer (PBS) was used as controls.

Effects of B. lanceolatus venom on vWF collagen (line 98): Suggestion for inclusion new results of more concentrations lower than used on it to determine the value for 50% of activity. 

Authors: Thank to reviewer’ comments, we have performed new series of experiments aimed to evaluate the effects of venom concentrations below 1 µg/mL on vWF to collagen type I , type III and type VI binding activities. Please find this new information in “Results” section and Figures of the revised manuscript

Figure 2: As suggestion the author could perform the experiment of B. lanceolatus effects on vWF collagen binding (high and low concentration of vWF) including venom concentration between 1 and 5 µg/mL to cover the gap of activity showed on it; it will be possible to go for no/less activity to 100% of activity.

Authors: As stated above, we have chosen to extend information regarding the effects of venom concentrations below 1 µg/mL on vWF to collagen type I , type III and type VI binding activities. We motivate this choice by the fact that venom plasma concentrations in patients bitten by Bothrops snakes are found around 10 to 100-1000 ng/mL.

Figure 3:  Please, improve the visual quality of this figure.

Authors: Figure visual quality has been improved. 

It is known that the conventional treatment for snake envenomation is performed according to the severity of bite (mild, moderate and severe). The author showed on Figure 3 two different concentrations of antivenom, which were able to inhibit the venom effect at 5µg/ml. However, they showed that these concentrations were inefficient when used the concentration 50µg/ml of venom. As suggestion, the authors should explore the effect difference according to venom concentration and connect the result obtained with the clinical use of antivenom, describing what is the usual dosage used for clinical treatment and efficiency percent showed on the paper. Thus, I highly suggest testing and show more than two concentration of snake venom for the inhibition assay with antivenom.

Authors: We agree with reviewer’ comments regarding the clinical use of antivenom. However, please understand that our study aim was to test whether B. lanceolatus venom may induce opposite effects on collagen type I and collagen type III versus collagen type VI vWF binding properties.

Grammar revision. There are many long sentence and word repetition on it.

Authors: The revised manuscript has been reviewed by a native English speaker in order to avoid long sentence and word repetition.

The authors should explore more information about the Bothrofav’s composition and the possible mechanism for venom inhibition, especially for the hemorrhagic effect.

Authors: Bothrofav’s composition has been provided in the revised manuscript. Please see the last paragraph of the “Discussion” section of the revised manuscript.

Sentence from line 271 – 273: please do a grammar revision, there are some incorrect verb conjugations on it.

Authors: The sentence has been improved according to reviewer’ comments.

Item “Determination of vWF collagen binding activity assay”: please check how the world vWF:CB was write, there are two different way for them.

Authors: The paragraph has been improved according to reviewer’ comments.

In general, the paper showed clearly an important aspect for Bothrops venom and will contribute for giving more information for futures studies on Toxinology. In conclusion, the improvement suggested will increase the paper quality for publication.

Authors: We thank the reviewer for its valuable comments. Improvements suggested by the reviewer have taken into account.

In general, the English language is clearly, however must be performed an review to improve the quality for publication

Authors: The revised manuscript has been reviewed by a native English speaker.

Round 2

Reviewer 1 Report

The authors did not carry out additional experiments to understand the alterations of the binding of vWF to collagens, such as SDS-PAGE. Simply using words as "increasing", "decreasing", or "inhibiting" the binding vWF to  collagen does not explain the results observed and the conclusions about thrombotic events in patients bitten by Bothrops lanceolatus. More detailed experiments are required, as I previously  commented.

Furthermore, the text keeps on showing typos and grammar errors. Revision is thoroughly required.

Poor usage of English language.

Author Response

Authors: We agree that this preliminary report did not answer the question regarding the molecular mechanism(s) leading to alterations of the binding of vWF to collagens. Nevertheless, to the best of our knowledge, the present work has the merit of being the first observation of the effects of B. lanceolatus venom on vWF to collagen binding activity, which seemingly contrast with the reported effects of other Bothrops spp.

Moving forward, we plan to further investigate the molecular mechanism(s) behind vWF to collagen binding activity (application of techniques such as SDS-PAGE, proteomics, western blotting, …).

Furthermore, we have taken into account the reviewer’s comment about the present work’s results being only able to generate hypotheses about the mechanisms behind thrombotic events in B. lanceolatus human envenomation. As such, we have carefully reviewed the manuscript.

Finally, please not that our co-author, Mrs. Rishika BANYDEEN, a native English speaker, has carefully reviewed the manuscript as per English language, writing style and grammar.

Reviewer 3 Report

The authors modified the new version including all suggestions made. They also answer properly all questions and explained the point that could not be changed. The new version presented was improved and it is in a good condition for publication. Considering the modifications and improvement I recommend the version for publication.  

Best regards.

Author Response

Authors: We thank the reviewer for his/her valuable comments.
